# Depressive symptoms and cardiovascular disease: a population-based study of older adults in rural Burkina Faso

Ben Brinkmann ![ORCID],[1] Collin F Payne,[2] Iliana Kohler,[3] Guy Harling ![ORCID],[4,5,6,7] Justine Davies,[8] Miles Witham ![ORCID],[9,10] Mark J Siedner ![ORCID],[11] Ali Sie,[12] Mamadou Bountogo,[12] Lucienne Ouermi,[12] Boubacar Coulibaly,[12] Till Bärnighausen[1,7]

► Prepublication history and supplemental material for this paper is available online. To view these files, please visit the journal online (http://dx.doi.org/10.1136/bmjopen-2020-038199).

For numbered affiliations see end of article.

**Correspondence to**
Mr Ben Brinkmann;
ben.brinkmann@stud.uni-heidelberg.de

## ABSTRACT

**Objectives** To contribute to the current understanding of depressive disorders in sub-Saharan African (SSA) countries by examining the association of depressive symptoms with cardiovascular and cardiometabolic conditions in a population-based study of middle-aged and older adults in rural Burkina Faso.

**Setting** This study was conducted in the Nouna Health and Demographic Surveillance System in northwestern Burkina Faso, in a mixed rural and small-town environment. The data were obtained between May and July 2018.

**Participants** Consenting adults over 40 years of age (n=3026).

**Primary and secondary outcome measures** Depressive symptoms were assessed using the Patient Health Questionnaire depression module (PHQ-9). Chronic cardiometabolic conditions were assessed via a lipid panel and glycated haemoglobin measures from serum, alongside anthropometry and blood pressure measurements and a self-reported questionnaire. Multivariable linear regression was used to test the relationship between depressive symptoms and cardiovascular/cardiometabolic conditions after controlling for sociodemographic factors.

**Results** Depressive symptoms were not associated with the metabolic syndrome (standardised beta coefficient=0.00 (95% CI −0.04 to 0.03)), hypertension (beta=0.01 (95% CI −0.02 to 0.05)), diabetes mellitus (beta=0.00 (95% CI −0.04 to 0.04)) and past diagnosis of elevated blood pressure or blood sugar. Prior stroke diagnosis (beta=0.04 (95% CI 0.01 to 0.07)) or heart disease (beta=0.08 (95% CI 0.05 to 0.11)) was positively associated with the standardised PHQ-9 score as were self-reported stroke symptoms.

**Conclusion** Objectively measured cardiometabolic conditions had no significant association with depressive symptoms in an older, poor, rural SSA population, in contrast to observations in high income countries. However, consequences of cardiovascular disease such as stroke and heart attack were associated with depressive symptoms in older adults in Burkina Faso.

## Strengths and limitations of this study

► This study was large and population based with a randomly selected sample in contrast to the selective nature of prior studies about depression in Burkina Faso.

► The use of biomarker testing for cardiovascular health and the Patient Health Questionnaire depression screening instrument that has been validated in primary care settings in sub-Saharan Africa provide internationally comparable data.

► The oral translation of the study questionnaire into local languages makes it vulnerable to differences in expression during interviews.

► The cross-sectional design of the questionnaire limits our ability to determine temporal ordering of and causal links between cardiovascular and mental health events.

► We were not able to clinically confirm diagnoses of hypertension, diabetes and depression in study participants.

## INTRODUCTION

The decline in communicable diseases and the ageing of populations worldwide, including low-income countries, means that cardiovascular disease (CVD) is now the leading cause of mortality worldwide, accounting for an estimated 31% of all deaths.[1] The leading cause of morbidity in the Global Burden of Disease study 2017, however, was depressive disorders.[2]

In high-income contexts, CVD risk factors as summarised by the metabolic syndrome—obesity, dyslipidaemia, hypertension and diabetes mellitus—are bidirectionally associated with depressive disorders.[3–6] People with depression are more likely to smoke tobacco, eat an unhealthy diet, be physically inactive and use alcohol to excess, which makes them more vulnerable to develop the metabolic

syndrome.[7] Elevated activity of the hypothalamus–pituitary–adrenal axis caused by chronic stress in people with depression increases the risk for obesity, dyslipidaemia and diabetes.[8 9] Alterations in the autonomic nervous system found in people with depression[10] contribute to hypertension and increased insulin resistance. Other related factors adding to the hypothesis of a shared pathophysiology are overactivated inflammatory pathways, oxidative stress, endothelial dysfunction and hormone disequilibrium.[5 11–13]

CVDs such as ischaemic heart disease and cerebrovascular stroke are associated with subsequent depression. While depression increases the risk of heart attack and stroke,[14 15] both are also associated with higher levels of later depression.[16–18] Poststroke depression might be a result of the brain damage from the stroke but can also be a psychological reaction to the stroke illness and to its disabling consequences. Risk factors for depressive symptoms after stroke include stroke severity, disability after stroke, cognitive impairment, lack of social support and genetic predisposition.[19 20] Furthermore, depression worsens the outcomes of stroke[20] and myocardial infarction,[21] increasing mortality and worsening long-term functional results.

In recent years, a small but growing body of evidence from sub-Saharan Africa (SSA) suggests that associations between CVD, diabetes and depression seen in high-income countries (HIC) may not generalise to lower income settings.[22 23] In addition, it has been found that the age pattern for depressive disorders differs substantially from HIC. Depression prevalence in HIC peaks around age 60 years and declines in the following decades, resulting in an inverted U-shape.[24] In contrast, in low-income contexts, the prevalence of depressive symptoms appears to increase consistently with age.[23 25]

In this paper, we seek to contribute to the current understanding of depressive disorders in SSA countries by using newly collected data from Burkina Faso. Specifically, we aim to examine the potential associations of depressive symptoms with the metabolic syndrome, hypertension, diabetes mellitus and self-reported chronic conditions and stroke symptoms as well as the age pattern of depressive symptoms. Depression and mental disorders more generally are understudied in Burkina Faso, and the evidence about the relationships with CVDs is inconclusive. Two recent studies examined the prevalence of depressive symptoms within preselected populations: Yaméogo et al[26] focused on hypertensive outpatients and Napon et al[27] investigated depressive symptoms among people who had a stroke. Duthé et al[28] worked with a sample of urban adults in the capital city of Ouagadougou, finding an association between broadly defined self-reported chronic conditions and depressive symptoms. Very recently, Ouédraogo et al[29] conducted a population-based study determining the prevalence of depressive symptoms in Burkina Faso for the first time.

In contrast to these previous studies, our work uses a large population-based sample in a mixed rural and small-town environment among a population of adults over 40 years of age. This allows us to examine the possible associations of depressive symptoms and CVD in older adults with less selection bias. It is of great importance to gather more information about the relationships between depressive symptoms and other diseases' morbidity in this SSA context as it can provide novel ideas and solutions to better serve the healthcare needs of these populations.

## METHODS
### Sample
This population-based study was conducted in the Nouna Health and Demographic Surveillance System (HDSS) in north-western Burkina Faso, which is run by the Centre de Recherche en Santé de Nouna.[30] This HDSS is located in a rural region consisting of 58 villages centred around the town of Nouna. In 2015, there were about 107 000 individuals within the area, living in approximately 15 000 households. Of these individuals, approximately 18 000 were over the age of 40 years; we selected 3998 of these individuals in a stratified two-stage cluster random sampling approach. We expected 25% non-response due to mortality, inadequate mobility or people rejecting to participate, with an aimed 3000 responses. Our first-stage sampling unit was at the level of villages and the seven sectors of Nouna town. Where there were fewer than 50 individuals aged over 40 years (six villages in the 2015 census), we included all of them in our sample. We then took the remaining sample in equal proportions from all remaining villages and sectors, selecting at random within village/sectors.

### Data collection
Between May and July 2018, all sampled individuals were invited to participate at their homes. Consenting participants completed a questionnaire on socioeconomic and demographic characteristics, their physical, cognitive and mental health, and access to and usage of healthcare. Field workers were trained to administer the questionnaire, using French as the main language. Translations from French to the local dialect Djula were done within the training module and clearly communicated to all field workers verbally. Written Djula literacy is very limited in this setting, necessitating verbal rather than written translation.

Indicators of physical health such as weight, height and blood pressure (BP) were measured by study staff. BP was measured using Omron Series 7 portable BP machines (Omron Healthcare, Kyoto, Japan). After 15 min of rest, three measurements were taken in the left arm of the seated respondents with 5 min between each measurement. The mean of the second and third measurement was calculated and used in analyses.

A blood sample was collected to measure glycated haemoglobin (HbA1c), triglycerides, high-density lipoprotein (HDL) and total cholesterol. To measure blood glucose levels, a point-of-care finger-prick test was carried

out. The participants were asked when they last consumed food or beverages to determine whether the glucose values were fasting or non-fasting. All blood collection as well as finger-prick tests were conducted by trained and certified phlebotomists.

## Outcome variable

To measure depressive symptoms, the Patient Health Questionnaire depression module (PHQ-9) was used. The PHQ-9 is an instrument designed for primary care, either to make a probable diagnosis of major depressive disorder (MDD) or to continuously measure depressive symptoms with a score between 0 and 27.[31] Higher scores represent more severe depressive symptoms. The scale consists of nine items, with participants asked if they experienced a range of depressive symptoms in the past 2 weeks, including: (1) little interest or pleasure in doing things; (2) feeling down, depressed or hopeless; (3) trouble falling or staying asleep, or sleeping too much; (4) feeling tired or having little energy; (5) poor appetite or overeating; (6) feeling bad about yourself; (7) trouble concentrating on things; (8) moving or speaking so slowly/rapidly that other people could have noticed; and (9) thoughts that you would be better off dead of or hurting yourself in some way. The response categories are: (0) not at all, (1) several days, (2) more than half of the days and (3) nearly every day.

Because PHQ-9 cut-points for diagnosing MDD in the SSA context remain unclear,[32] we use the PHQ-9 as a continuous score to reflect the amount of depressive symptoms an individual experienced in the past 2 weeks prior to the questioning. We emphasise that the score does not equal a diagnosis of depression. This linear specification avoids a binary categorisation and allows subthreshold depression to be evaluated in our analyses. Subthreshold depression is characterised by clinically relevant symptomatology, and while less severe than MDD, its high prevalence can generate greater economic burden; in higher income settings, it is responsible for more doctor–patient consultations than MDD.[33 34] To account for potential interviewer effects, we calculated Z-scores for the PHQ-9 results within interviewer before analysis (online supplemental figures 1 and 2).

## Explanatory variables

Based on the Harmonized Joint Scientific Statement (Harmonized) criteria,[35] we defined someone as having the metabolic syndrome if they met at least three of five conditions: (1) body mass index (BMI) ≥30 kg/m$^2$; (2) mean systolic BP ≥130 mm Hg or a mean diastolic BP ≥85 mm Hg; (3) triglycerides ≥150 mg/dL; (4) HDL cholesterol <40 mg/dL in men and <50 mg/dL in women; and (5) fasting plasma glucose ≥100 mg/dL.

We also considered several individual chronic conditions as explanatory variables: having a positive screening result for hypertension grade II (ie, BP levels of a mean systolic BP ≥140 mm Hg or a mean diastolic BP ≥90 mm Hg)[36]; having a positive screening result for prediabetes,

defined as a fasting plasma glucose value between 100 and 126 mg/dL and/or an HbA1c value between 5.7% and 6.5%, based on the classification criteria of the American Diabetes Associations[37]; and having a positive screening result for diabetes mellitus, defined as a fasting plasma glucose value ≥126 mg/dL or an HbA1c value ≥6.5%.

In the remainder of this article, we define 'hypertension' as a positive screening result for hypertension, rather than the clinically diagnosed condition (ie, by a medically trained healthcare provider) according to country-specific requirements. The same accounts for 'diabetes mellitus' and the 'metabolic syndrome'.

Last, we used self-reported measures of chronic health conditions. These included whether the respondent reported ever being diagnosed with: elevated BP; elevated blood sugar; a heart attack; a stroke; or had ever experienced any of three stroke symptoms: sudden drooping of one side of the face; sudden numbness, weakness or dead feeling on one half of the body; or sudden difficulty speaking or slurring of speech. The stroke symptoms were selected according to the FAST ('Face', 'Arm', 'Speech' and 'Time') system, which was created by the Brain Attack Coalition based on the Cincinnati Pre-Hospital Stroke Scale in 1999.[38] We also created a 'self-reported metabolic syndrome' variable, defined as reporting any of the first four self-reported diagnoses.

The wealth index was constructed using polychoric principal components analysis (PCA) methods as described by Kolenikov and Angeles.[39] This method is similar to the Filmer and Pritchett[40] PCA method commonly used by Demographic and Household Surveys[41] but incorporates continuous variables and a lack of ownership into wealth index quantification. Variables in the final calculation included, among others: main source of water, toilet type, number of bedrooms, number of cows, horses and other animals, hectares of land, electricity, cell phone and bank account.

## Statistical analysis

Baseline descriptive statistics were obtained and means calculated. Confidence Intervals (CI) for binomial variables were calculated using the Clopper-Pearson intervals. Multivariable linear regression was conducted to see how physical morbidity measures were associated with depressive symptoms. The standardised beta coefficients that are used to express the degree as to which morbidity variables are associated with the PHQ-9 score represent the percent value in which the probability of a yes/no event changes for every one standard deviation (SD) change in the PHQ-9 score. All models were adjusted for demographic and socioeconomic characteristics of the respondents, including age (in 5-year groups), sex, place of residence (rural/semirural), marital status (not married/married), education (none/some, but none completed/primary completed or more), ethnicity (Dafin, Mossi, Bwama and other) and a wealth index. We standardised all coefficients so that they represented the SD of change in the PHQ-9 score associated with a one-unit change in

morbidity variable. Missing values were excluded listwise from analyses. All analyses were conducted using SPSS V.25 (IBM, New York, USA).

### Patients and public involvement

Patients and the public were not involved in the design, or conduct, or reporting, or dissemination plans of our research.

## RESULTS

Summary statistics for the sample used in the present analysis are presented in table 1. Of the 3998 sampled individuals, 3026 (75.7%) participants were found, consented and completed all questions from the PHQ-9 depression module and were included in the analysis. A total of 1523 (50.3%) of the respondents were women and 1503 were men. Women were older on average (mean: 55.5 years, SD: 11.2) than men (mean: 53.1 years, SD: 10.6). They were also more likely to be without formal schooling (90.7%) than men (78.1%), less likely to be married (55.3%, men: 84.7%) and had a higher mean BMI (22.5, SD: 5.0; men: 21.8, SD: 3.7). Hypertension, diabetes and hypertriglyceridaemia all were equally distributed between both sexes. Elevated BP was highly prevalent in the sample: 27.3% screened positive for grade II hypertension. The prevalence of diabetes in this population was relatively low: 6.8%. Among the women, 70.8% had an HDL cholesterol level below the cut-point to fulfil the criteria for the metabolic syndrome, while only 45.7% of the men had an HDL cholesterol below the cut-off.

### Age patterns of depressive symptoms

As figure 1 shows, the absolute mean values of the Z-scored PHQ-9 scores rose continually with age in both men and women. Considering only men, the mean PHQ-9 value for the youngest group of respondents, the 40–44 years old, was −0.41 (95% CI −0.49 to −0.34). This value rose constantly up to 0.88 (95% CI 0.55 to 1.2) within the oldest age group of over 75 years old. PHQ-9 scores were generally higher among women corresponding to higher levels of depressive symptoms that were also positively correlated with age. The mean value for women in the youngest age group was −0.16 (95% CI −0.25 to −0.07), rising to 0.80 (95% CI 0.54 to 1.05) for women in the oldest group. Underlying data can be found in supplemental materials (online supplemental table 1).

### Age patterns of the metabolic syndrome, diabetes mellitus and hypertension

Figure 2 shows that the prevalence of the metabolic syndrome increased in a statistically significant way with age for women: it increased from a mean of 12.0% (95% CI 8.3% to 16.6%) within 40–44 years old to 28.2% (95% CI 18.6% to 39.5%) within the 75+ years age group. Surprisingly, the prevalence rate for men declined slightly with age, going from 8.2% (95% CI 5.4% to 11.8%) to 5.7% (95% CI 0.0% to 15.7%), though this difference

was not statistically significant. A small increase in the prevalence of prediabetes and diabetes was observed with increasing age, but this difference was not statistically significant. While an increasing prevalence with age could not be shown for diabetes mellitus, we did find a positive association between age and hypertension. Both men and women showed a rising prevalence of elevated BP with increasing age. Underlying data can be found in supplemental materials (online supplemental tables 2-5).

### Association of depressive symptoms with the metabolic syndrome, its components, hypertension and diabetes mellitus

The first panel of table 2 and figure 3 show the standardised coefficients from simple linear regressions of the Z-scored PHQ-9 score on the metabolic syndrome, its components, hypertension and diabetes. In all models, higher age and being female were strongly associated with a higher PHQ-9 score. Having the metabolic syndrome, being positively screened for hypertension, prediabetes or diabetes was not associated with a higher PHQ-9 score. Neither was any of the single components of the metabolic syndrome.

### Association of depressive symptoms with self-reported conditions and stroke symptoms

Figure 3 shows that having previously been diagnosed by a healthcare worker with elevated BP or elevated blood sugar was not significantly associated with the PHQ-9 score, whereas the models did show a positive association for ever having been diagnosed with a stroke (standardised beta coefficient (hereafter 'beta')=0.04 (95% CI 0.01 to 0.07)) or heart disease (beta=0.08 (95% CI 0.05 to 0.11)). An even stronger association could be shown for reporting to ever having had a sudden drooping of one side of the face (beta=0.09 (95% CI 0.06 to 0.12)), a sudden numbness weakness or dead feeling on one half of the body (beta=0.14 (95% CI 0.10 to 0.17)) and a sudden slurring of speech (beta=0.14 (95% CI 0.11 to 0.18)).

The 'self-reported metabolic syndrome' with one or more out of four self-reported health items present was associated positively with the PHQ-9 score (beta=0.07 (95% CI 0.04 to 0.10)). The results did not change substantially when running the regression models for the self-reported health items within the parts of the population who were screened positively for the respective condition, for example, the model with self-reported elevated BP within the population who were measured as hypertensive (online supplemental tables 23-25).

## DISCUSSION

Depressive symptoms were not associated with the metabolic syndrome, hypertension, diabetes mellitus and past diagnosis of elevated BP or blood sugar. Prior stroke diagnosis or heart disease were positively associated with the standardised PHQ-9 score, as were self-reported stroke

**Table 1** Sample characteristics

| | Total (%) | Female (%) | Male (%) |
|---|---|---|---|
| | n=3026 (100) | n=1523 (50.3) | n=1503 (49.7) |
| Age group (years) | | | |
| 40–44 | 685 (22.6) | 300 (19.7) | 385 (25.6) |
| 45–49 | 579 (19.1) | 250 (16.4) | 329 (21.9) |
| 50–54 | 478 (15.8) | 252 (16.5) | 226 (15.0) |
| 55–59 | 393 (13.0) | 202 (13.3) | 191 (12.7) |
| 60–64 | 304 (10.0) | 178 (11.7) | 126 (8.4) |
| 65–69 | 249 (8.2) | 142 (9.3) | 107 (7.1) |
| 70–74 | 166 (5.5) | 94 (6.2) | 72 (4.8) |
| ≥75 | 172 (5.7) | 105 (6.9) | 67 (4.5) |
| Place of residence | | | |
| Nouna | 838 (27.7) | 452 (29.7) | 386 (25.7) |
| Village | 2188 (72.3) | 1071 (70.3) | 1117 (74.3) |
| Ethnicity | | | |
| Dafin | 1176 (38.9) | 575 (37.8) | 601 (40.0) |
| Bwama | 927 (30.6) | 490 (32.2) | 437 (29.1) |
| Mossi | 403 (13.3) | 199 (13.1) | 204 (13.6) |
| Other* | 520 (17.1) | 259 (16.9) | 261 (17.3) |
| Education | | | |
| No formal schooling | 2555 (84.4) | 1381 (90.7) | 1174 (78.1) |
| Some schooling but none completed | 253 (8.4) | 79 (5.2) | 174 (11.6) |
| Primary schooling completed and more | 218 (7.2) | 63 (4.1) | 155 (10.3) |
| Currently married | 2115 (69.9) | 842 (55.3) | 1273 (84.7) |
| Body mass index† | | | |
| <18.5 | 498 (16.8) | 268 (17.8) | 230 (15.6) |
| 18.5–24.9 | 1884 (63.4) | 875 (58.3) | 1009 (68.8) |
| 25.0–29.9 | 441 (14.8) | 249 (16.6) | 192 (13.1) |
| ≥30.0 | 148 (5.0) | 110 (7.3) | 38 (2.6) |
| Hypertension‡ | | | |
| Grade I | 1674 (56.3) | 841 (56.0) | 833 (56.7) |
| Grade II | 812 (27.3) | 433 (28.8) | 379 (25.8) |
| Diabetes mellitus§ | | | |
| Prediabetes | 974 (45.4) | 530 (46.6) | 444 (44.1) |
| Diabetes | 139 (6.8) | 84 (7.7) | 55 (5.7) |
| Hypertriglyceridaemia¶ | 335 (11.6) | 177 (12.0) | 158 (11.2) |
| Low HDL** cholesterol†† | 1366 (58.6) | 849 (70.8) | 517 (45.7) |
| Metabolic syndrome‡‡ | 394 (15.6) | 260 (20.4) | 134 (10.7) |
| Self-reported conditions | | | |
| Elevated blood pressure | 507 (17.1) | 294 (19.8) | 213 (14.4) |
| Elevated blood sugar | 70 (2.3) | 32 (2.1) | 38 (2.6) |
| Heart disease | 166 (5.5) | 116 (7.7) | 50 (3.3) |
| Stroke | 40 (1.3) | 16 (1.1) | 24 (1.6) |
| Self-reported stroke symptoms | | | |
| Sudden drooping | 58 (1.9) | 33 (2.2) | 25 (1.7) |
| Sudden numbness | 240 (7.9) | 133 (8.7) | 107 (7.1) |

Continued

**Table 1** Continued

| | Total (%) n=3026 (100) | Female (%) n=1523 (50.3) | Male (%) n=1503 (49.7) |
|---|---|---|---|
| Sudden slurring | 159 (5.3) | 88 (5.8) | 71 (4.7) |
| Self-reported metabolic syndrome§§ | 688 (23.3) | 396 (26.8) | 292 (19.9) |

*Peulh, Samo and other.

†Calculated by weight in kilograms divided by squared height in metres (kg/m²), 55 missing values due to missing height/weight measurements.

‡Hypertension grade I: mean systolic blood pressure (BP) ≥130 mm Hg or mean diastolic BP ≥85 mm Hg; hypertension grade II: mean systolic BP ≥140 mm Hg or mean diastolic BP ≥90 mm Hg. 55 missing values due to missing BP measurements.

§Prediabetes: fasting plasma glucose (FPG) value between 100 and 126 mg/dL and/or an HbA1c value between 5.7% and 6.5%, 882 missing values due to missing HbA1c measurements and non-fasting results that were excluded. Diabetes: FPG value ≥126 mg/dL or an HbA1c value ≥6.5 %, 982 missing values due to missing HbA1c measurements and non-fasting results that were excluded.

¶Hypertriglyceridaemia: triglycerides ≥150 mg/dL, 135 missing values due to missing triglyceride measurements.

**High-density lipoprotein cholesterol.

††Low HDL cholesterol: <40 mg/dL in women, <50 mg/dL in men, 696 missing values due to missing HDL cholesterol measurements.

‡‡502 metabolic syndrome: three of five conditions: (1) body mass index ≥30 kg/m²; (2) mean systolic BP ≥130 mm Hg or mean diastolic BP ≥85 mm Hg; (3) triglycerides ≥150 mg/dL; (4) HDL cholesterol <40 mg/dL in men, <50 mg/dL in women; or (5) FPG ≥100 mg/dL. Missing values due to missing height, weight, HDL cholesterol and triglyceride measurements as well as non-fasting glucose measurements.

§§78 missing values.

HbA1c, glycated haemoglobin; HDL, high-density lipoprotein.

symptoms. Additionally, the prevalence of depressive symptoms rose substantially with increasing age for both men and women, confirming similar findings from recent studies investigating depression in SSA and standing in contrast to findings from HICs. The findings indicate that in a setting where cardiovascular conditions are not the dominant health issue, non-symptomatic conditions like low-grade hypertension do not have detrimental effects on people's mental health.

### Chronic conditions and depressive symptoms

The findings suggest that the metabolic syndrome, hypertension and diabetes have no significant association with depressive symptoms in our sample. This situation is congruent with recent studies from SSA investigating the relationship between CVD and depression. Gelaye et al[42] did not find an association between depression and the metabolic syndrome or diabetes in a sample of

Ethiopian adults, and Geldsetzer et al[22] showed that there was no association between depression and hypertension, diabetes and obesity in a South African sample. In contrast to these results from SSA, relationships between CVD and depression have been widely reported in HIC.[3–6 43] In SSA, non-communicable diseases including CVD are only recently emerging as a public health concern, while infectious diseases are still the most important cause of death.[44] People and health systems are more attuned to seeing and treating acute infections and maternal health problems

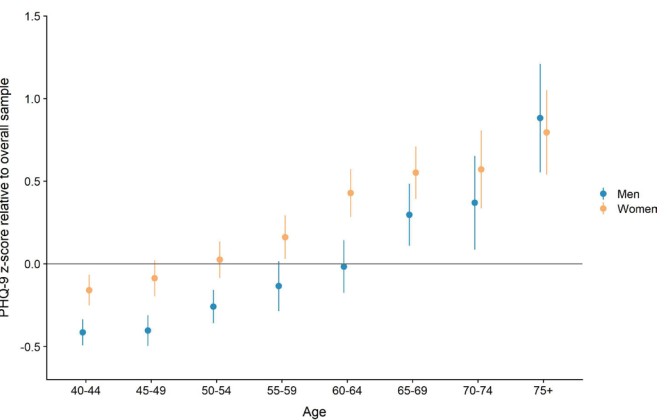

**Figure 1** Mean PHQ-9 scores (Z-scored within interviewer) and 95% CIs. PHQ-9, Patient Health Questionnaire depression module. Underlying datacan be found in supplemental materials (online supplemental table 21).

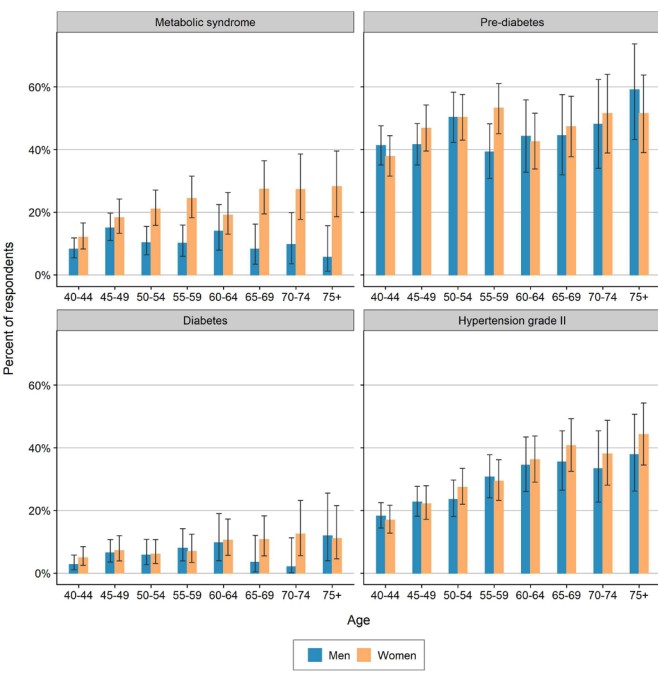

**Figure 2** Prevalence of the metabolic syndrome, prediabetes, diabetes and hypertension; 95% CIs. Underlying data can be found in supplementary materials (online supplemental tables 2-5).

**Table 2** Standardised beta coefficients derived from separate multivariable linear regression models investigating the relationship between cardiometabolic and cardiovascular conditions and the Z-scored PHQ-9 results

| | Standardised beta coefficient | 95% CI | P value |
|---|---|---|---|
| Metabolic syndrome | −0.004 | (−0.040 to 0.033) | 0.847 |
| Obesity | 0.007 | (−0.028 to 0.042) | 0.680 |
| Hypertriglyceridaemia | −0.006 | (−0.040 to 0.029) | 0.742 |
| Reduced HDL cholesterol | −0.027 | (−0.067 to 0.012) | 0.177 |
| Hypertension grade I | 0.028 | (−0.007 to 0.062) | 0.113 |
| FPG >100 | −0.022 | (−0.061 to 0.019) | 0.298 |
| Hypertension grade II | 0.014 | (−0.020 to 0.048) | 0.427 |
| Prediabetes | 0.005 | (−0.034 to 0.043) | 0.819 |
| Diabetes | −0.001 | (−0.040 to 0.039) | 0.972 |
| Self-reported conditions | | | |
| Hypertension | 0.032 | (−0.003 to 0.066) | 0.071 |
| Diabetes | 0.004 | (−0.029 to 0.038) | 0.809 |
| Heart disease | 0.081 | (0.047 to 0.114) | <0.001 |
| Stroke | 0.038 | (0.005 to 0.071) | 0.024 |
| Self-reported stroke symptoms | | | |
| Sudden drooping | 0.090 | (0.057 to 0.123) | <0.001 |
| Sudden numbness | 0.136 | (0.102 to 0.168) | <0.001 |
| Sudden slurring | 0.144 | (0.110 to 0.176) | <0.001 |
| Self-reported metabolic syndrome | 0.070 | (0.035 to 0.104) | <0.001 |

Each standardised coefficient arises from a separate model controlling for age, sex, gender, place ofresidence, education, ethnicity and wealth quintile. The separate models can be found in (online supplemental tables 6-22).
FPG, fasting plasma glucose; HDL, high-density lipoprotein; PHQ-9, Patient Health Questionnaire depression module.

than CVD, which might lead to a lower degree of understanding of what it means to have CVD, hence protecting the mental health of people in SSA. This hypothesis will need to be tested in future studies. However, it is known that individuals with a higher socioeconomic status (SES) can afford higher healthcare costs than those with a lower SES, for whom it is a greater burden to pay for their health. Fiske *et al*[17] found that unaffordable healthcare

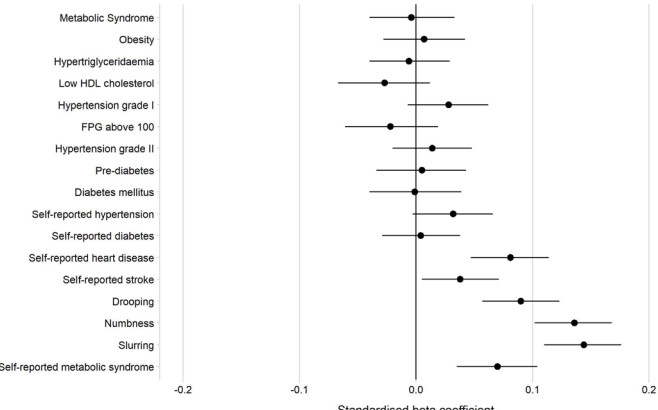

**Figure 3** Forest plot of associations between non-communicable conditions and PHQ-9 Z-scores. FPG, fasting plasma glucose; HDL, high-density lipoprotein; PHQ-9, Patient Health Questionnaire depression module.

costs for individuals affected by CVD may have a negative effect on their mental health.

CVD risk factors are asymptomatic, especially in their early stages, which might reduce the negative effects of CVD on mental health. Research in HIC has shown that there are biological factors like chronic inflammation, vascular endothelial dysfunction, neuroendocrine disequilibrium (particularly of the pituitary–adrenal axis) and increased platelet activation that can cause depression in people with CVD and vice versa.[45 46] It remains unclear in the SSA context if these biomedical processes are similarly important in driving an association between depressive symptoms and CVD; further research is needed to investigate this question. Multimorbidity might be another factor linking CVD and depression in HIC.[47] According to a recent study conducted in the same HDSS as this study, multimorbidity is highly prevalent in this sample.[48] Whether experiencing other non-communicable diseases in addition to CVD—for instance cancer, chronic obstructive pulminary disorder (COPD), musculoskeletal disorders and other mental illnesses—has an effect on the relationship between depressive symptoms and CVD has not been examined and could be subject to further investigations. Use of antidepressants is linked to a higher risk of obesity, diabetes and CVD[49] and could thus act as a confounding factor in studies

in HIC investigating the association between depressive symptoms and cardiometabolic risk. In Burkina Faso, the availability of specialised care is poor[50] and so is the availability of antidepressants, which makes it less likely for Burkinabé to experience the adverse outcomes of antidepressants.

## Consequences of CVD

Interestingly, our findings suggest that having experienced the consequences of a stroke as well as having ever been diagnosed with a stroke or heart disease is associated with higher PHQ-9 results. These associations are in congruence with the situation in HICs, where especially ischaemic events like heart attacks and strokes due to CVD are associated with depression.[45] Ischaemic events may lead to physical dysfunction, impairing the ability to work in physical jobs that are important for the social and economic livelihood in a rural SSA environment. Social insecurity and disintegration resulting from these limitations are substantial risk factors to develop depressive symptoms. According to Schatz and Seeley,[51] older adults in SSA strongly link their physical functionality to the concept of seeing themselves as independent individuals. With declines in physical functioning, their perceived independence deteriorates, leaving individuals as increasingly dependent and at risk of depression. Limited economic resources hinder the ability of older adults in rural areas to cope with stressors like disability after stroke and also act to limit access to an already weak health system. This paucity of diagnosis and treatment options also acts as part of a vicious cycle, as it may lead to substantial delays in accessing the healthcare system.[52] These barriers to mental healthcare lead to a lack of knowledge and awareness of an individual's own physical and mental health, making it much less likely to engage in protective behaviours such as regular physical activity, avoiding stimulants and drugs, engaging in stabile relationships or adjusting sleep rhythms and diet.[53]

## Sociodemographics

Women had a higher prevalence of depressive symptoms than men in our sample, showing that it will be especially important to target women in future interventions. With increasing age, the prevalence of depressive symptoms rose substantially for both men and women, confirming similar findings from recent studies investigating depression in SSA.[22 23] Mechanisms that protect older populations in HIC include an increasing and positive influence of religiosity, gathered wisdom and life experience, higher economic resources, intensified social engagement and socioemotional selectivity.[17 18] Examining whether these resilience mechanisms apply to the SSA context should be a focus for future research.

## Implications

The findings suggest that improving care for measurable CVD conditions—metabolic syndrome, hypertension and diabetes—may have additional benefits for improving depressive symptoms. Even though they might not be directly associated with depressive symptoms, these conditions increase the likelihood of thromboembolic events that are associated with worse mental health. When depression is detected in its early stages, it might also be possible to prevent a part of CVD burden by engaging in preventive diagnostics and treatment in a targeted part of the population. While it is highly prevalent and likely an important correlate and basis for other chronic conditions and care success, depression has been woefully neglected in the global health response. As diagnostic and therapeutic options exist, this gap should be addressed by policymakers as soon as possible.

## Strengths and limitations

Our study had several strengths. It was large and population-based with a randomly selected sample, in contrast to the selective nature of almost all prior studies about depression in Burkina Faso, with the notable exception of Ouédraogo *et al*.[29] The use of both questionnaire and biomarker testing allowed objective measurement of chronic conditions, using internationally standardised values allowing comparison with studies elsewhere. Furthermore, the PHQ-9 instrument is locally applicable, being widely used in population-based studies and primary care settings in SSA.[23 54–56] Nevertheless, its translation into the local language, particularly the collaborative manner in which it was done, makes it vulnerable to differences in expression during interviews. While we have adjusted for differential understanding by standardising scores at the interviewer level, cultural differences in expressing psychological states like sadness may still be important. The cross-sectional design of the questionnaire and the one-time sample do not make it possible to draw causal interpretations from the findings or to confirm diagnoses of hypertension, diabetes and depression. Finally, our sample of adults aged 40 years and above is not directly comparable with the typically older cut-offs for ageing studies; however, given the life expectancy of Burkinabé people (60.8 years[57]), we believe that the respondents represent old individuals within this setting. Clearly our findings only apply to over 40s, and associations may differ in younger individuals.

Our analyses do not account for the potential that some individuals in our sample may be on medication to control a cardiovascular condition. Respondents who self-reported having hypertension, elevated blood sugar, hypercholesterolaemia or heart disease were asked a single combined question on whether they had received any treatment in the last 2 weeks. However, the definition of 'treatment' was not specified any further and therefore may represent treatments as varied as medication, physical exercise, dietary change or traditional healing. Self-reported rates of treatment were generally low, with only 11% of those with hypertension and 33% of those with self-reported hypertension reported having received treatment in the last 2 weeks. The numbers for elevated blood sugar were even lower: 9% of those with diabetes

and 23% of those with self-reported elevated blood sugar said they had received treatment in the last 2 weeks. Our data showing low treatment rates are consistent with studies in SSA that document similar patterns for hypertension[58–61] and diabetes.[62] The non-specific nature of the available treatment variable means that any analyses using this variable would be nearly impossible to meaningfully interpret, and we therefore chose to omit self-reported treatment from our analyses.

## CONCLUSION

In conclusion, we find that having the metabolic syndrome, hypertension or diabetes was not associated with the results of the PHQ-9 score among a group of older adults in a rural, SSA setting. Interestingly, an association could be found between depressive symptoms and having experienced stroke symptoms as well as self-reported diagnose of stroke and heart disease. This might be due to the influence of disability after thromboembolic events on the psychological well-being of African adults seen in prior research. It may also be seen as a development foreshadowing an association of CVD and depression in this population, which cannot be seen yet because of lacking awareness, diagnosis and treatment options. Age was strongly associated with depressive symptoms, confirming the age patterns found in previous studies in SSA.

**Author affiliations**
[1]Heidelberg Institute of Global Health, Heidelberg University, Heidelberg, Germany
[2]School of Demography, The Australian National University, Canberra, Australian Capital Territory, Australia
[3]Population Studies Center (PSC) and Department of Sociology, University of Pennsylvania, Philadelphia, Pennsylvania, USA
[4]Institute for Global Health, University College London, London, UK
[5]Harvard Center for Population and Development Studies, Harvard T.H. Chan School of Public Health, Cambridge, Massachusetts, USA
[6]MRC/Wits Rural Public Health & Health Transitions Research Unit (Agincourt), University of the Witwatersrand, Johannesburg, South Africa
[7]Africa Health Research Institute, KwaZulu-Natal, South Africa
[8]Institute of Applied Health Research, University of Birmingham, Birmingham, UK
[9]AGE Research Group, NIHR Newcastle Biomedical Research Centre, Newcastle University, Newcastle upon Tyne, UK
[10]Newcastle Upon Tyne Hospitals NHS Foundation Trust, Newcastle Upon Tyne, UK
[11]Department of Medicine, Massachusetts General Hospital, Harvard Medical School, Boston, Massachusetts, USA
[12]Centre de Recherche en Sante de Nouna, Nouna, Boucle du Mouhoun, Burkina Faso

**Contributors** TB and GH conceived and designed the overall Centre de Recherche en Santé de Nouna (CRSN) Heidelberg Aging Study (CHAS) study. CFP, IK, JD, MW and MJS contributed to the design of the CRSN CHAS household survey. BB, GH, AS, MB, LO and BC coordinated baseline data collection and preparation. BB conducted the analysis, and wrote and revised the manuscript. CFP and IK supervised the analysis, write up and development of the manuscript. All authors substantively reviewed manuscripts, inputted into revisions and approved the final manuscript.

**Funding** Funding for the CRSN Heidelberg Aging Study (CHAS) was provided by the Alexander von Humboldt Foundation through an Alexander von Humboldt Professor award to TB, which is funded by the German Federal Ministry of Education and Research. MW acknowledges support from National Institute for Health Research Newcastle Biomedical Research Centre. GH is supported by a fellowship from the Wellcome Trust and Royal Society (210479/Z/18/Z). CFP acknowledges support from the ANU Futures Scheme.

**Disclaimer** The funder has no role in the design of the study and collection, analysis and interpretation of data and in writing the manuscript of the study.

**Competing interests** None declared.

**Patient consent for publication** Not required.

**Ethics approval** This study received ethics approval from the national health ethics committee in Ouagadougou (#2018-5-053), the CRSN institutional ethics committee in Nouna (#2018–04) and the ethics committee of the medical faculty of the Ruprecht-Karls-Universität Heidelberg (#S-120/2018).

**Provenance and peer review** Not commissioned; externally peer reviewed.

**Data availability statement** Data are available on reasonable request. Data are not publicly available as consent was not given by participants for data to be shared openly. This is in part because entire age cohorts of some villages are included in the dataset, potentially allowing for deductive disclosure with sufficient local information. For this reason, anonymised data are available from CHAS study data controllers only following signature of a data use agreement restricting onward transmission. Anyone wishing to replicate the analyses presented, or conduct further collaborative analyses using CHAS (which are welcomed and considered based on a letter of intent), should contact Dr Guy Harling (g.harling@ucl.ac.uk) in the first instance.

**ORCID iDs**
Ben Brinkmann http://orcid.org/0000-0002-1726-5430
Guy Harling http://orcid.org/0000-0001-6604-491X
Miles Witham http://orcid.org/0000-0002-1967-0990
Mark J Siedner http://orcid.org/0000-0003-3506-842X

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
