## [Reviewer comments · BMJ Open]

ARTICLE DETAILS

TITLE (PROVISIONAL)	Depressive symptoms and cardiovascular disease: a population-based study of older adults in rural Burkina Faso
AUTHORS	Brinkmann, Ben; Payne, Collin F.; Kohler, Iliana; Harling, Guy; Davies, Justine; Witham, Miles; Siedner, Mark; Sie, Ali; Bountogo, Mamadou; Ouermi, Lucienne; Coulibaly, Boubacar; Bärnighausen, Till

VERSION 1 – REVIEW

REVIEWER	Liqiang Zheng Shengjing hospital of China Medical University, China
REVIEW RETURNED	17-Mar-2020

GENERAL COMMENTS	Depressive symptoms and cardiovascular disease: a population-based study of older adults in rural Burkina Faso The objectives of this article are to contribute to the current understanding of depressive disorders in sub-Saharan African (SSA) countries by examining the association of depressive symptoms with cardiovascular and cardiometabolic conditions in a population-based study of middle-aged and older adults in rural Burkina Faso. The authors finding that depressive symptoms were not associated with the metabolic syndrome, hypertension, diabetes mellitus, past diagnosis of elevated blood pressure or blood sugar. And prior stroke diagnosis, heart disease or self-reported stroke symptoms were positively associated with the standardized PHQ-9 score. The overall analysis of the article is specific, clear and fully discussed. Some suggestions and questions are as follows: Major comments : 1.The article is designed as cross-sectional and lacks the clinical diagnosis of the corresponding disease, which limits the determination of the causal link between depression and related diseases.2.This article explores the relationship between depressive symptoms and diseases such as metabolic syndrome, hypertension, diabetes, and stroke, but the medication status of the affected population, such as antihypertensive drugs, hypoglycemic drugs, and antiplatelet drugs, may have an important effect on the results, but did not see the relevant drug status, the author should analyze. Minor comments: 1.The abstract on page 4 shows that the data was collected from May to June 2018, and the 36th row on page 9 is written from May to July 2018. Please unify.2.Introduction Line 11 on page 7, GBD 2015 is cited, and GBD 2017 has been updated, please update the reference.
--

	3. On page 8, line 47, "between depressive symptoms and other morbidity in this SSA context" is suggested to be changed to "other diseases' morbidity". 4. Method section on page 9 suggests adding a sphygmomanometer type. 5. On line 10 of the outcome variable section on page 10, it is recommended to add "the higher the score, the more severe" after "with a score between 0 and 27". 6. Definition of hypertension grade II (mean systolic BP \geq 140 mmHg or a mean diastolic BP \geq 90 mmHg) on page 11, line 16, proposed added the 2017 U.S. guidelines for hypertension. 7. On page 11, line 40, what are the basis that three symptoms of the stroke that the author chose? 8. Line 18 of the results section on page 13, "Of the 3,998 sampled individuals", line 17 of the sample section on page 9 reads "we selected 4,000 of these individuals"? 9. Continuous variables (such as age) of the results, please add standard deviation. 10. Is the definition of hypertriglyceridemia consistent with the criteria in the Metabolic Syndrome section? If not, please provide a definition. 11. The relevant definitions in the article table please explain in the remarks. 12. Discussion section suggests adding the first paragraph of the main findings of this article.
--	---

REVIEWER	Andrzej Pająk Jagiellonian University Medical College
REVIEW RETURNED	02-Apr-2020

GENERAL COMMENTS	General comment Despite important limitations in data collection (including that forms were not translated to the local language in writing but translated from the French version on the spot of examination by the interviewers) and in the interpretation of the results, the study may be of some interest for international society. It indicates that in the society in which CVD is not a dominant health problem, depression is not related to health problems which are not related to patient's subjective recognition of the patient, like CVD risk factors but in contrast it is strongly related to those which are recognized. Finding that metabolic syndrome and even diabetes are not related with depressive symptoms but awareness of metabolic syndrome seems to be in line of the above. Minor queries 1) Authors should pay more attention to make tables and figures self-understood by making titles more descriptive and giving descriptions for the axis in each case. 2) It is not understood why descriptive statistics for total cholesterol were not given in the table 1.
---

VERSION 1 – AUTHOR RESPONSE

Reviewer Comments, Author Responses and Manuscript Changes

Reviewer: 1

Reviewer Name: Liqiang Zheng

Institution and Country: Shengjing hospital of China Medical University, China
Please state any competing interests or state 'None declared': None

Dear Dr. Zheng,

thank you very much for your detailed and thoughtful review. Your comments were very helpful for the revision of our manuscript and helped improved it. Below please find our responses to your comments and suggestions.

Please leave your comments for the authors below

Depressive symptoms and cardiovascular disease: a population-based study of older adults in rural Burkina Faso

The objectives of this article are to contribute to the current understanding of depressive disorders in sub-Saharan African (SSA) countries by examining the association of depressive symptoms with cardiovascular and cardiometabolic conditions in a population-based study of middle-aged and older adults in rural Burkina Faso. The authors finding that depressive symptoms were not associated with the metabolic syndrome, hypertension, diabetes mellitus, past diagnosis of elevated blood pressure or blood sugar. And prior stroke diagnosis, heart disease or self-reported stroke symptoms were positively associated with the standardized PHQ-9 score. The overall analysis of the article is specific, clear and fully discussed.

Some suggestions and questions are as follows:

Major comments:

1. The article is designed as cross-sectional and lacks the clinical diagnosis of the corresponding disease, which limits the determination of the causal link between depression and related diseases.

Correct, this is a cross-sectional study and we estimate associations rather than causal relationships. To clarify this, we have added text discussing this aspect and the lack of clinical diagnosis in (1) the "strengths and limitations" section on page 4 lines 21-22 and (2) the methods section page 10 lines 7-16 where we refer to the conditions "hypertension/diabetes" as either "positive screening result for hypertension/diabetes" or "elevated blood pressure/sugar". A similar approach was used for the metabolic syndrome, which we referred to as "positive screening result for the metabolic syndrome". Hypertriglyceridemia, reduced HDL-cholesterol and elevated fasting plasma glucose are descriptive conditions derived solely from blood values.

We included the sentence "The cross-sectional design of the questionnaire limits our ability to determine temporal ordering of and causal links between cardiovascular and mental health events." in the "strengths and limitations" section page 4 lines 19-20, with the newly added "and causal links between".

Additionally, we added the following clarification: "In the remainder of this article, we define 'hypertension' as a positive screening result for hypertension, rather than the clinically diagnosed condition (i.e., by a medically trained health care provider) according to country specific requirements. The same accounts for 'diabetes mellitus' and the 'metabolic syndrome'." (page 10 lines 21-23)

As a consequence, we refer to “a positive screening result for hypertension/diabetes/metabolic syndrome” as “hypertension/diabetes/metabolic syndrome” throughout the document.

2. This article explores the relationship between depressive symptoms and diseases such as metabolic syndrome, hypertension, diabetes, and stroke, but the medication status of the affected population, such as antihypertensive drugs, hypoglycemic drugs, and antiplatelet drugs, may have an important effect on the results, but did not see the relevant drug status, the author should analyze.

We included the following statement in the strengths and limitations section (page 20 line 22 – page 21 line 6):

“Our analyses do not account for the potential that some individuals in our sample may be on medication to control a cardiovascular condition. Respondents who self-reported having hypertension, elevated blood sugar, hypercholesterolemia, or heart disease were asked a single combined question on whether they had received any treatment in the last two weeks. However, the definition of “treatment” was not specified any further, and therefore may represent treatments as varied as medication, physical exercise, dietary change, or traditional healing. Self-reported rates of treatment were generally low, with only 11% of those with hypertension and 33% of those with self-reported hypertension reported having received treatment in the last two weeks. The numbers for elevated blood sugar were even lower: 8.6% of those with diabetes and 22.9% of those with self-reported elevated blood sugar said they had received treatment in the last two weeks. Our data showing low treatment rates are consistent with studies in SSA that document similar patterns for hypertension and diabetes. The non-specific nature of the available treatment variable means that any analyses using this variable would be nearly impossible to meaningfully interpret, and we therefore chose to omit self-reported treatment from our analyses.”

Minor comments:

1. The abstract on page 4 shows that the data was collected from May to June 2018, and the 36th row on page 9 is written from May to July 2018. Please unify.

Good catch, thanks for pointing out this discrepancy. The data was collected between May and July 2018 and we corrected this throughout the manuscript.

2. Introduction Line 11 on page 7, GBD 2015 is cited, and GBD 2017 has been updated, please update the reference.

The reference has been updated.

3. On page 8, line 47, "between depressive symptoms and other morbidity in this SSA context" is suggested to be changed to "other diseases' morbidity".

The sentence has been changed accordingly.

4. Method section on page 9 suggests adding a sphygmomanometer type.

We included the following statement in the methods section page 8 lines 27-30:

“Blood pressure was measured using Omron Series 7 portable blood pressure machines (Omron Healthcare, Kyoto, Japan). After 15 minutes of rest, three measurements were taken in the left arm of the seated respondents with 5 minutes between each measurement. The mean of the second and third measurement was calculated and used in analyses.”

5. On line 10 of the outcome variable section on page 10, it is recommended to add "the higher the score, the more severe" after "with a score between 0 and 27".

The following sentence has been added at the suggested position in the text on page 9 line 10 of the outcome variable section:

“Higher scores represent more severe depressive symptoms.”

6. Definition of hypertension grade II (mean systolic BP \geq 140 mmHg or a mean diastolic BP \geq 90 mmHg) on page 11, line 16, proposed added the 2017 U.S. guidelines for hypertension.

The reference has been added.

7. On page 11, line 40, what are the basis that three symptoms of the stroke that the author chose?

The following explanation has been added to the methods section page 10 line 28:

“The stroke symptoms were selected according to the FAST (“Face”, “Arm”, “Speech”, “Time”) system, which was created by the Brain Attack Coalition based on the Cincinnati Pre-Hospital Stroke Scale in 1999.”

8. Line 18 of the results section on page 13, "Of the 3,998 sampled individuals", line 17 of the sample section on page 9 reads "we selected 4,000 of these individuals"?

3,998 is the correct number of sampled individuals. It has been corrected in the sample section page 8 line 8.

9. Continuous variables (such as age) of the results, please add standard deviation.

Standard deviation added for age (page 12 line 20) and BMI (page 12 line 22).

10. Is the definition of hypertriglyceridemia consistent with the criteria in the Metabolic Syndrome section? If not, please provide a definition.

Yes, the definition for hypertriglyceridemia is a blood value of triglycerides \geq 150 mg/dl for both the single condition and the criteria in metabolic syndrome.

11. The relevant definitions in the article table please explain in the remarks.

The definitions of each condition are now explained in the footnotes of Table 1.

12. Discussion section suggests adding the first paragraph of the main findings of this article.

The following paragraph has been added as a summary of the main findings (page 16 line 24 to page 17 line 2):

“Depressive symptoms were not associated with the metabolic syndrome, hypertension, diabetes mellitus and past diagnosis of elevated blood pressure or blood sugar. Prior stroke diagnosis or heart disease were positively associated with the standardized PHQ-9 score, as were self-reported stroke symptoms. Additionally, the prevalence of depressive symptoms rose substantially with increasing age for both men and women, confirming similar findings from recent studies investigating depression in SSA and standing in contrast to findings from high income countries. The findings indicate that in a setting where cardiovascular conditions are not the dominant health issue, non-symptomatic conditions like low-grade hypertension do not have detrimental effects on people’s mental health.”

Reviewer: 2

Reviewer Name: Andrzej Pająk

Institution and Country: Jagiellonian University Medical College

Please state any competing interests or state ‘None declared’: No competing interests

Dear Dr. Pająk,

thank you very much for your very helpful comments on our manuscript. Below please find our responses.

Please leave your comments for the authors below

General comment

Despite important limitations in data collection (including that forms were not translated to the local language in writing but translated from the French version on the spot of examination by the interviewers) and in the interpretation of the results, the study may be of some interest for international society. It indicates that in the society in which CVD is not a dominant health problem, depression is not related to health problems which are not related to patient's subjective recognition of the patient, like CVD risk factors but in contrast it is strongly related to those which are recognized. Finding that metabolic syndrome and even diabetes are not related with depressive symptoms but awareness of metabolic syndrome seems to be in line of the above.

Minor queries:

1. Authors should pay more attention to make tables and figures self-understood by making titles more descriptive and giving descriptions for the axis in each case.

The title of Table 2 has been extended. Also, descriptions have been added for all axes in figures 1-3.

2. It is not understood why descriptive statistics for total cholesterol were not given in the table 1.

Total cholesterol did not enter the analyses at any point. We used HDL-cholesterol because of the possible inclusion in the criteria for the metabolic syndrome.

VERSION 2 – REVIEW

REVIEWER	Liqiang Zheng Shengjing Hospital of China Medical University
REVIEW RETURNED	08-May-2020
GENERAL COMMENTS	The author has answered all questions, I have no more questions
REVIEWER	Andrzej Pająk Jagiellonian University, Medical College
REVIEW RETURNED	13-May-2020
GENERAL COMMENTS	No special comments